# Remote Sensing Monitoring of Grassland Locust Density Based on Machine Learning

**DOI:** 10.3390/s24103121

**Published:** 2024-05-14

**Authors:** Qiang Du, Zhiguo Wang, Pingping Huang, Yongguang Zhai, Xiangli Yang, Shuai Ma

**Affiliations:** 1College of Information Engineering, Inner Mongolia University of Technology, Hohhot 010080, China; 20211800119@imut.edu.cn (Q.D.);; 2Inner Mongolia Key Laboratory of Radar Technology and Application, Hohhot 010051, China; 3College of Information Science and Engineering, Chongqing Jiaotong University, Chongqing 400074, China

**Keywords:** grassland locust, machine learning, regression, remote sensing

## Abstract

The main aim of this study was to utilize remote sensing data to establish regression models through machine learning to predict locust density in the upcoming year. First, a dataset for monitoring grassland locust density was constructed based on meteorological data and multi-source remote sensing data in the study area. Subsequently, an SVR (support vector regression) model, BP neural network regression model, random forest regression model, BP neural network regression model with the PCA (principal component analysis), and deep belief network regression model were built on the dataset. The experimental results show that the random forest regression model had the best prediction performance among the five models. Specifically, the model achieved a coefficient of determination (R^2^) of 0.9685 and a root mean square error (RMSE) of 1.0144 on the test set, which were the optimal values achieved among all the models tested. Finally, the locust density in the study area for 2023 was predicted and, by comparing the predicted results with actual measured data, it was found that the prediction accuracy was high. This is of great significance for local grassland ecological management, disaster warning, scientific decision-making support, scientific research progress, and sustainable agricultural development.

## 1. Introduction

Grasslands are a crucial ecosystem in China, serving not only as a significant geographic barrier but also as the country’s primary natural ecological defense line. Grasslands play a vital role in maintaining ecological balance and biodiversity. However, grassland locust plagues not only severely hinder the growth of grassland vegetation and the development of local pastoralism, but also bring substantial economic losses to local herders, affecting the healthy development of the region’s pastoralism and grassland ecology [1]. Locusts primarily feed on Poaceae plants, such as wheat, rice, maize, and various pastures; leguminous and Cyperaceae plants; and some vegetables [2]. In the Inner Mongolia grasslands, the main locust species causing significant environmental damage include the Asian migratory locust, the short-winged locust, Acrida cinerea, and the large-winged locust. These locust populations have a significant negative impact on ecological balance. Locusts require exposed ground surfaces for egg-laying. In extensive pastoral and agro-pastoral areas, due to improper grassland management and excessive grazing pressure, the overgrazing and degradation of grasslands occur, creating favorable conditions for the large-scale breeding of grassland locusts. Furthermore, grassland pest infestations exacerbate the degradation and desertification of grasslands. Combined with drought, reduced rainfall, land exposure, and a reduction in natural predators, these factors collectively contribute to a vicious cycle [3].

With the advancement of remote sensing technology, the methods used to monitor grassland locusts have shifted from traditional, time-consuming, and less accurate ground surveys to more efficient remote sensing techniques. Remote sensing technology can rapidly collect vast amounts of ground information and produce various remote sensing data products for surface parameters, providing strong support for studying the habitats of grassland locusts. However, current locust monitoring methods based on single environmental factors often overlook other factors that affect locust growth. Therefore, these methods have limited spatial universality and poor transferability across regions. In addition to studying the mechanism of habitat and locust occurrence using single factors, research institutions and scholars both domestically and internationally have begun to consider multiple habitat factors for locust monitoring. Unless causing devastating damage, locust activity is usually not evident in satellite remote sensing data; therefore, the remote sensing monitoring of locusts almost always adopts an indirect method: indirectly assessing the occurrence of locusts by monitoring their habitats. This method focuses on analyzing and understanding the environmental conditions that locusts rely on, rather than directly observing the locusts themselves [4].

However, current studies have not fully exploited the potential of multi-source remote sensing data and their long-term sequences in the study of locust disasters. At the same time, they have not fully tapped into the value of historical changes in habitat factors. To address these issues, this study explored the potential of using long-term multi-source remote sensing data to predict grassland locust density in breeding areas. By combining machine learning techniques, a pest prediction model with strong spatial universality and high temporal stability was constructed, enabling the prediction of the potential risks of grassland locusts on a large scale. This provides an important reference value for the prevention and control of grassland locust disasters [5].

## 2. Materials and Methods

### 2.1. Study Area

Xiwuzhumuqin Banner covers a total area of 2,245,938 hectares, of which the grassland area reaches up to 2,213,200 hectares. The available grassland area is 2,029,000 hectares, accounting for 88.2% of the total area. The cultivated land area in this region is 187.09 hectares, and the sandy land area is 27,990.91 hectares. The grasslands in Xiwuzhumuqin Banner are divided into 5 major categories, 11 subcategories, and 88 types. In terms of surface water resources, there are 7 major rivers in the region, belonging to the Wulagai River system, mainly flowing from south to north. The total length of these rivers is 1789 km, and the watershed area is 2,296,000 hectares [6].

Climatically, Xiwuzhumuqin Banner is located in the mid-latitude inland area and has a mid-temperate arid and semi-arid continental climate. Under this climate, the spring is windy and prone to drought, the summer is warm with uneven rainfall, the autumn is cool with early frost and snow, and the winter is cold and long. Xiwuzhumuqin Banner is located at the junction of North China and Northeast China, and is a typical grassland pastoral area.

The average annual precipitation in this region is 350 mm, showing a decreasing trend from southeast to northwest. The annual average temperature is 1.2 °C, with the extreme highest temperature reaching 37.4 °C and the lowest temperature dropping to −38.6 °C. The average frost-free period is 105 days, the average number of days with strong winds (above force 7) is 62 days, and the average sunshine duration is 2900 h.

Xiwuzhumuqin Banner has diverse grassland types, including mountainous meadow steppes, low-mountain hilly meadow steppes, semi-desert steppes, and river–floodplain and lake basin lowland meadow steppes. The shrub and forestland area in this region is 74,500 hectares. There are 14 rivers in the territory, with 7 major rivers, 326 lakes, and 60 mountain springs [7]. The climatic conditions in the study area are suitable for the survival and reproduction of grassland locusts, so these areas have long been disaster-prone zones for grassland locust infestations. The region mainly relies on agriculture and animal husbandry, so the disasters caused by grassland locusts have a significant impact on the economic development of these study areas. Figure 1 demonstrates the specific location of the study area, which is located in Xiwuzhumuqin Banner, XilinGol League, Inner Mongolia Autonomous Region, China.

### 2.2. Grassland Locust Density Data

The sample grasshopper data used in this study were obtained from the Xiwuzhumuqin Banner Grassland Workstation. As a local forestry and grassland management unit, this institution is responsible for compiling annual data on grasshopper density. Grasshopper disaster survey data are collected from mid-May to early June each year, as the early control of grasshoppers in this region is crucial to ensure the normal operation of animal husbandry and protect it from the damage caused by grasshopper infestations.

The first step in the grasshopper density survey was to select the survey area, which was set at one square kilometer. Seventy percent of the areas with dense grasshopper activity and thirty percent of the areas with sparse grasshopper activity were selected to form a control group. The second step was to select survey sites, randomly and evenly choosing a certain number of sites within the survey area. The third step was sampling, using a one square meter enclosed container to cover the sampling points, ensuring no gaps between the container and the ground that would allow the grasshoppers to escape. Insecticide was then sprayed into the container and, after the grasshoppers died, they were collected and counted. The fourth step was statistics, where the number of grasshoppers at each sampling point was divided by the number of sampling points to represent the average grasshopper density in the survey area. The central latitude and longitude of the survey area correspond to the latitude and longitude of the grasshopper density value.

The survey areas cover Xiwuzhumuqin Banner, following the standard pest survey procedures of the forestry and grassland department. This study collected grasshopper disaster data from 2021 and 2022, for a total of 160 sampling points, with 80 for each year. Figure 2 shows the locations of the grasshopper survey sites in 2021 and 2022.

### 2.3. Meteorological Data

The meteorological data used in this study were sourced from the Xiwuzhumqin Banner Meteorological Station. These data have a temporal resolution of one ten-day period, covering a range of metrics, including average temperature, precipitation, surface temperature, and soil moisture. The data span from January 2020 to December 2022. The meteorological data from the station were primarily used to double-check the remote sensing meteorological data, aiming to enhance the accuracy and reliability of the overall dataset.

### 2.4. Multi-Source Remote Sensing Data

The daily 1 km all-weather land surface temperature dataset of China’s mainland and surrounding areas has a temporal resolution of four times per day and a spatial resolution of 1 km. Data from 2020 to 2022 were selected, covering the spatial scope of China. The method used to prepare the dataset was the enhanced satellite thermal infrared remote sensing–reanalysis data integration method. The main input data of the method were Terra/Aqua MODIS LST products and GLDAS data, and the auxiliary data included the vegetation index and surface albedo provided by satellite remote sensing. The method fully utilized the high-frequency components, low-frequency components, and spatial correlation of land surface temperature provided by satellite thermal infrared remote sensing and reanalysis data and, finally, it reconstructed a high-quality all-weather land surface temperature dataset [8]. The dataset can be downloaded from the following website: https://data.tpdc.ac.cn/en/data/05d6e569-6d4b-43c0-96aa-5584484259f0/ (accessed on 18 February 2024).

The daily all-weather surface soil moisture dataset of China has a 1 km resolution (2003–2022) and was generated by downscaling the SSM (surface soil moisture), based on AMSR-E (Advanced Microwave Scanning Radiometer for EOS) and AMSR-2 (Advanced Microwave Scanning Radiometer 2) data, from a 36 km resolution to a 1 km resolution, significantly surpassing the well-known combined SMAP/Sentinel (active-passive microwave) SSM product at a 1 km resolution. It boasts a temporal resolution of 1 day and a spatial resolution of 1 km [9]. Data from 2020 to 2022 were downloaded. The dataset can be downloaded from the following website: https://data.tpdc.ac.cn/en/data/e1f24e35-6235-40b2-b3d7-677dfb249e39/ (accessed on 18 February 2024).

The Monthly Precipitation Dataset of China with a Resolution of 1 km under Multiple Scenarios and Modes for 2021–2100 is a dataset that collects monthly precipitation data in China under multiple scenarios and modes. The spatial resolution of this dataset is 0.0083333° (approximately 1 km), and the data selected covered the period from January 2020 to December 2022. The data are in NETCDF format. This dataset was generated by downscaling the global climate model dataset with a resolution of >100 km released by the IPCC Coupled Model Intercomparison Project Phase 6 (CMIP6) and the global high-resolution climate dataset published by WorldClim using the delta spatial downscaling scheme in China. The geospatial scope of the dataset covered the main land areas of China [10]. The dataset can be downloaded from the following website: https://data.tpdc.ac.cn/zh-hans/data/a9cd4a09-51a9-433b-9540-0376c6134cf6 (accessed on 18 February 2024).

The MYD13Q1 dataset is a part of MODIS (Moderate-Resolution Imaging Spectroradiometer) and is a global vegetation index (NDVI) product. This dataset provides important information about the status of surface vegetation. Global MYD13Q1 data are provided every 16 days with a spatial resolution of 250 m. The data were atmospherically corrected, removing interference caused by clouds, heavy aerosols, and cloud shadows. The dataset has reached validation stage 3, indicating that its quality and reliability have been rigorously evaluated by the scientific community. MOD13Q1 data are the same. Combined, the MOD13Q1 and MYD13Q1 datasets form a dataset with a time resolution of 8 days, and they come from NASA’s Terra and Aqua satellites, respectively, which have slightly different orbits and observation times, but both provide vegetation index updates every 16 days. By reasonably combining these two datasets, the temporal resolution could be increased, allowing for the more frequent monitoring of vegetation changes. We also selected data from 2020 to 2022.

The aforementioned downloaded remote sensing data covered the period from January 2020 to December 2022, representing a significant amount of data. Therefore, Figure 3 only shows random 1-day remote sensing data maps of the study area for several types of remote sensing data, including soil moisture data, precipitation data, land surface temperature data, and NDVI data.

Table 1 below shows the acquisition time of the data used in this study.

### 2.5. Correlation Analysis between Meteorological Factors and Locust Density

This study took the locust density as the target dependent variable. At the same time, a Pearson correlation analysis was conducted on the original environmental variables, including soil moisture, daytime land surface temperature, the Normalized Difference Vegetation Index, cumulative precipitation, and night-time land surface temperature. In addition, the “random forest–Gini importance” (RF GI) method was employed to rank the importance of environmental factors. By combining these two methods, the selection of input variables was achieved, and the characteristic variables of the input dataset were determined.

Taking the correlation analysis between locust density in 2021 and land surface temperature from 1 January 2020 to 30 December 2021, as an example, Table 2 shows the correlation between locust density and 10-day average (daytime/night-time) land surface temperature, as well as the confidence level of this correlation.

Taking the correlation analysis between locust density in 2021 and precipitation from 1 January 2020 to 30 December 2021, as an example, Table 3 displays the correlation between locust density and average precipitation in 10-day periods, along with the confidence level of this correlation.

Taking the correlation analysis between locust density in 2021 and soil moisture from 1 January 2020 to 30 December 2021, as an example, Table 4 displays the correlation between locust density and average soil moisture in 10-day periods, as well as the confidence level of this correlation.

Taking the correlation analysis between locust density in 2021 and NDVI from 1 January 2020 to 30 December 2021, as an example, Table 5 displays the correlation between locust density and average NDVI in 10-day periods, along with the confidence level of this correlation.

Taking the correlation analysis between locust density in 2021 and various meteorological factors from 1 January 2020 to 30 December 2021, as an example, Table 6 displays the random forest importance scores of environmental factor variables relative to locust density. Through the establishment of a random forest model, we analyzed the importance of each environmental factor at each 10-day time point relative to the locust density data. Only some of the higher-scoring data are presented in Table 6. The remaining data, which have too low scores, are not shown in the table.

Specifically, the random forest constructs multiple decision trees and makes predictions by averaging or voting on these trees. During the construction of each tree, the algorithm considers all features and calculates the information gain of each feature at the splitting nodes. Information gain measures the reduction in uncertainty or entropy of the sample response (in this case, locust density) after splitting using that feature. Features with a higher information gain are considered more important for model prediction.

Therefore, in random forest regression, by calculating the average information gain of each feature across all trees, we could obtain the importance score of that feature in the model. These scores help us understand which environmental factor variables are the most critical for predicting locust density. In the random forest model, “gain” typically refers to the average value of the information gain of a feature (i.e., an environmental factor variable) when used to split samples during tree construction. Information gain is a metric for measuring feature importance, indicating how much the purity of the dataset (i.e., the degree of clustering of samples from the same class) has improved after splitting based on that feature. Thus, in Table 6, “gain” can be understood as “information gain” or simply “gain,” representing the contribution of each environmental factor variable to improving the accuracy of predicting locust density in the random forest model.

Similarly, the correlation analysis between the locust density data and various meteorological factors in 2022 revealed the same correlation characteristics as in 2021. Finally, several important characteristic habitat factors were identified for the inversion of locust density: the average daytime land surface temperature in the last 10 days of February of the current year; the average daytime land surface temperature in the first 10 days of April of the current year; the average daytime land surface temperature in the middle 10 days of May of the current year; the average nighttime land surface temperature in the middle 10 days of August of the previous year; the average night-time land surface temperature in the middle 10 days of January of the current year; the average precipitation in the first 10 days of December of the previous year; the average precipitation in the first 10 days of April of the same year; the average precipitation in the middle 10 days of June of the same year; the average soil moisture in the first 10 days of July of the previous year; the average soil moisture in the middle 10 days of October of the previous year; the average soil moisture in the middle 10 days of April of the current year; the average NDVI in the middle 10 days of August of the previous year; and the average NDVI in the middle 10 days of May of the current year.

### 2.6. Deviation Normalization

Calculating the deviation normalization helps to eliminate the impacts of different units or scales when dealing with data. Since the distribution of environmental data is not normal or contains outliers, deviation normalization may be more suitable. This scales all values to a range of 0 to 1, and is more sensitive to outliers. For models such as neural networks, the deviation normalization operation helps to accelerate training and improve model performance because it maintains the distribution of the data. The formula for calculating deviation normalization is shown in formula (1):(1)$$Z=\frac{X_{i}-X_{\min}}{X_{\max}-X_{\min}}$$

In the formula, Z denotes the calculated value of the Min-Max normalization of the current environmental variable, $X_{i}$ denotes the current environmental variable at the time of operation, $X_{min}$ denotes the minimum value of the current environmental variable at the time of operation, and $X_{max}$ denotes the maximum value of the current environmental variable at the time of operation [11].

When analyzing the relationship between locust density and meteorological conditions, various meteorological variables, such as temperature and humidity, are involved. These variables have different units and ranges of values. Deviation normalization can convert all these variables to a unified scale (from 0 to 1), which helps to avoid certain variables dominating the model due to their larger numerical ranges [12].

## 3. Machine Learning-Based Grassland Locust Monitoring Model

### 3.1. Construction of the Dataset

The method used to construct the dataset for this study was as follows: after preprocessing the original data, missing values were filled in, and the temporal and spatial resolutions were unified. Based on the work conclusion of the previous section, the characteristic meteorological factor data used for inversing the locust density were obtained; the same feature data corresponding to different sample points were normalized via deviation, and a dataset was formed, as shown in Table 7.

In Table 7, “Serial Number” refers to the serial numbers of 180 locust density survey sites in 2021 and 2022. “N0_01” indicates the average Normalized Difference Vegetation Index (NDVI) in the middle 10 days of August of the previous year. “T_01” represents the average soil moisture in the middle 10 days of April of the current year. “D1_01” stands for the average daytime land surface temperature in the middle 10 days of May of the current year. “D2_01” means the average night-time land surface temperature in the middle 10 days of August of the previous year. “J_01” signifies the average precipitation in the middle 10 days of June of the same year. “N0_02” indicates the average NDVI in the middle 10 days of May of the current year. “HCMD” represents the average density of locusts at the sample site.

### 3.2. Locust Density Inversion Model

This study divided the dataset into training and test sets, and it constructed models based on BP neural network regression combined with the principal component analysis (PCA), random forest regression, BP neural network regression only, deep belief network regression, and support vector regression (SVR). Subsequently, the models underwent training and parameter optimization [13].

BP Neural Network Regression Based on Principal Component Analysis: The principal component analysis (PCA) has become a common method for handling high-dimensional data and simplifying datasets. The core purpose of this technique is to transform complex multi-dimensional data into a lower-dimensional subspace, while minimizing the overall loss of information in order to more effectively represent the original dataset. In meteorological data analyses, the PCA is particularly important, as factors such as rainfall, air humidity, and soil moisture often have close inter-relationships. By applying the PCA to transform these interrelated data, their dimensions can be reduced, thereby improving the efficiency of model training [14].

The BP (backpropagation) neural network, inspired by the human brain’s response mechanism, is a type of multi-layer, fully connected network primarily used for data fitting and classification [15]. It consists of three key components: the input layer, hidden layers, and the output layer. Neurons, as the fundamental units of the network, facilitate signal transmission between these layers. With the help of internal activation functions in the neurons, the BP neural network can approximate a variety of complex non-linear functions. The workflow of the BP neural network is as follows: signals propagate forward from the input layer, passing through multiple hidden layers, where the signal undergoes complex processing before reaching the output layer [16]. The data at the output layer are compared with the target data, generating an error value. If the current weights and thresholds do not produce the desired output, the error information will propagate back along the same path; that is, it backpropagates to each corresponding neuron, adjusting the weights and thresholds. This process repeats until the network output error falls within an acceptable range, completing the training process [17]. A model of the BP neural network is illustrated in Figure 4.

The calculation formula for the nodes in the hidden layer in the diagram is as follows:(2)$$y_{i}=\varphi (\sum_{i=1}^{n}\omega_{i}x_{i}+b_{i})$$

In formula (2), *n* represents the number of nodes; φ is the activation function, $\omega_{i}$ denotes the parameter weights for the *i*-th layer, and $b_{i}$ is the bias for the *i*-th layer. Combining the PCA and BP neural network for a regression analysis helps to reduce the risk of overfitting and enhances generalization to unseen data by eliminating noise and irrelevant variables from the data. However, it also comes with disadvantages [18]. The dimensionality reduction process may discard some components that are crucial for prediction, leading to a deterioration in the interpretability of the model. Combining these two techniques also implies the need to adjust and optimize more parameters, potentially complicating the model training and optimization processes.

Random Forest Regression: The random forest regression algorithm employs an ensemble method consisting of numerous independently constructed decision trees. The core process of this algorithm includes the following: firstly, the generation of multiple different training samples and attribute subsets by repeatedly sampling the original dataset with replacement; secondly, the construction of a decision tree for each sample and attribute subset; and, finally, the derivation of the final prediction value by voting or taking the weighted average of the predictions from these decision trees [19]. Compared with other machine learning techniques, a significant advantage of a random forest is its ensemble learning characteristic. A random forest can usually avoid the overfitting problem that might occur in a single decision tree, thereby improving generalizability to new data, as well as possessing good noise resistance. Moreover, a random forest maintains an efficient training speed, even when handling large datasets; it can process high-dimensional data without the need for feature selection; and it can provide assessments of the impact of each feature on prediction results, offering some basis for model interpretation [20]. In this study, the model used cross-validation. A schematic diagram of the random forest regression model is shown in Figure 5.

BP Neural Network Regression: In this model, BP neural network regression is used independently, without the implementation of the principal component analysis. BP neural networks are capable of capturing and modeling complex non-linear relationships, which is extremely valuable for complex datasets that are difficult to handle with linear models. BP neural networks can effectively predict unseen data, demonstrating good generalization capabilities.

Deep Belief Network Regression: Deep belief networks (DBNs) are a type of deep learning model composed of multiple layers of generative models; specifically, typically stacked Restricted Boltzmann Machines (RBMs). Each RBM layer learns representations of data at different levels of abstraction. DBNs initially employ unsupervised learning for the layer-wise pre-training of the network, followed by fine-tuning through supervised learning. DBNs are capable of automatically learning complex and high-level feature representations of data, which is particularly important in fields such as image and speech recognition [21]. DBNs generally demonstrate good generalization performance across a variety of tasks. Figure 6 shows a structural diagram of a deep belief network (DBN) model. This model includes three stacked RBM layers and one BP layer. DBNs initially conduct preliminary pre-training of the network through multiple RBM layers and utilize the BP layer for fine-tuning with supervised learning, thereby achieving comprehensive training of the model [22].

In this study, the RBM receives the data vector transmitted from the bottommost layer through the visible layer. The input vector undergoes an activation function transformation to the hidden layer and, through training, the internal energy function is minimized [23]. Given visible units $v_{i}$, hidden units $h_{j}$, and their connection weights $W_{i,j}$ (with a size of $n_{v}$,${\;n}_{h}$), as well as the offset $a_{i}$ for $v_{i}$ and the bias weight $b_{j}$ for $h_{j}$, the energy function E(v,h) is defined using formula (3):(3)$$\begin{array}{c} E\left(v,h\right)=\sum_{i=1}^{n_{v}}a_{i}v_{i}+\sum_{j=1}^{n_{h}}b_{j}h_{j}+\sum_{i=1}^{n_{v}}\sum_{j=1}^{n_{h}}h_{j}W_{i,j}v_{i} \end{array}$$

By calculating the energy function $E\left(v,h\right)$, the probability distribution P(v,h) for the visible and hidden layers can be expressed as Equations (4) and (5), where Z denotes the normalization factor:(4)$$\begin{array}{c} Z=\sum_{v}\sum_{h}e^{-E\left(v,h\right)} \end{array}$$
(5)$$\begin{array}{c} P\left(v,h\right)=e^{-\frac{E\left(v,h\right)}{Z}} \end{array}$$

The probability distribution $P_{\theta }(v)$, for observed data v, corresponding to the marginal distribution of $P_{\theta }(v,h)$, is referred to as the likelihood function, as shown in Equation (6). Equation (7) represents the vector obtained by removing component $h_{k}$ from h, and it is substituted into Equations (8) and (9).
(6)$$\begin{array}{c} P_{\theta }\left(v\right)=\sum_{h}P_{\theta }\left(v,h\right)=\frac{1}{Z_{\theta }}\sum_{h}e^{{-E}_{\theta }\left(v,h\right)} \end{array}$$
(7)$$\begin{array}{c} h_{-k}={\left(h_{1},h_{2},\cdots ,h_{k-1},h_{k+1},\cdots ,h_{nh}\right)}^{T} \end{array}$$
(8)$$\begin{array}{c} a_{k}\left(v\right)=b_{k}+\sum_{i=1}^{n_{v}}W_{k,i}v_{i} \end{array}$$
(9)$$\beta \left(v,h_{-k}\right)=\sum_{i=1}^{n_{v}}a_{i}v_{i}+\sum_{\begin{array}{c} j=1\\ i\neq k \end{array}}^{n_{h}}b_{j}h_{j}+\sum_{i=1}^{n_{v}}\sum_{\begin{array}{c} j=1\\ i\neq k \end{array}}^{n_{h}}h_{j}W_{j,i}v_{i}$$

The energy function simplifies to Equation (10), and the solution for the likelihood function is derived as shown in Equations (11) and (12):(10)$$\begin{array}{c} E\left(v,h\right)=-\beta \left(v,h_{-k}\right)-h_{k}\alpha_{k}\left(v\right) \end{array}$$
(11)$$\begin{array}{c} P\left(h_{k}=1|v\right)=P\left(h_{k}=1|h_{-k},v\right)=\frac{1}{1+e^{-\alpha_{k}\left(v\right)}}=sigmoid\left(\alpha_{k}\left(v\right)\right) \end{array}$$
(12)$$\begin{array}{c} P\left(v_{k}=1|h\right)=sigmoid\left(a_{k}+\sum_{j=1}^{n_{k}}W_{j,k}h_{j}\right) \end{array}$$

The activation probability formula for Restricted Boltzmann Machines (RBMs) is the sigmoid function. This function yields values between 0 and 1 for the entire range of (−∞, +∞), allowing for the computation of activation probabilities for respective nodes. When the activation status of all neural units in the visible layer (or hidden layer) is known, the activation probabilities for the hidden layer (or visible layer) neurons can be inferred. This involves calculating $P\left(h_{k}=1|v\right)$ and $P\left(v_{k}=1|h\right)$. The unknown RBM parameters W, a, and b can be determined through unsupervised learning [24].

SVR Model: Support vector regression (SVR) is a regression method based on support vector machines (SVMs). In traditional SVMs, the goal is to find a decision boundary that maximizes the margin between different classes of data points. In SVR, this concept is applied to regression problems, i.e., predicting a continuous value, rather than classification [25]. SVR allows for the setting of an “epsilon margin” within the model, which defines the acceptable error between predicted values and actual values. This approach helps to control the model’s generalization ability and the risk of overfitting. SVR is robust against outliers and noise [26]. The model primarily relies on support vectors (i.e., data points near the boundary) rather than all data, making it less sensitive to outliers [27]. SVR can effectively handle data in high-dimensional feature spaces, working well even when the number of features exceeds the number of samples [28].

## 4. Results and Discussion

### 4.1. Evaluation Criteria

In this study, BP neural network regression combined with the principal component analysis, random forest regression, BP neural network regression only, deep belief network regression, and SVR models based on the principal component analysis were applied to build models and compare their performance on grassland locust monitoring data. Inputting the habitat characterization dataset resulted in the prediction of grassland locust density in 2021 using the above five models. Subsequently, these predicted values were compared with the actual values in the test set and analyzed using scatter plots. The horizontal coordinates of the scatterplot represent the actual locust density in the test set, and the vertical coordinates represent the predicted values of the models. The diagonal line in the plot is a 1:1 line, indicating the exact agreement between the predicted and actual values. The closer the sample points are to the 1:1 line, the smaller the difference between predicted and actual values and, thus, the better the model prediction performance. If the predicted value is higher than the actual value, it will be above the 1:1 line; if the predicted value is lower than the actual value, it will be below the 1:1 line. When validating the effectiveness of a model, we often adopt two indicators: the coefficient of determination (R²) and the root mean square error (RMSE). The coefficient of determination (R²) is a key indicator to measure how well the regression model fits the sample data. The closer the value of R² to 1, the better the model fits the data, which means that the model can better explain the variation in the data.

### 4.2. Discussion of Results

Figure 7 below shows the results of the BP neural network regression combined with the principal component analysis. It can be seen that the coefficient of determination is 0.8718, which is a relatively high R² value, implying that the model’s predictions are of good quality, explaining most of the data variance. However, the overall performance is poor, and the predictions are not as good as expected. The root mean square error (RMSE) of 2.0476 indicates that the model’s predictions statistically deviate from the actual observations by an average of about 2.0476 units. The scatterplot shows that most of the data points are distributed close to or around the ideal line, indicating that the predicted values are close to the actual values. Data points in the range of actual values of 45 to 60 seem to have a better predictive accuracy because these points are more compactly distributed around the ideal line. For actual values exceeding 60, the predicted values appear to slightly overestimate the actual results, as most data points in the scatterplot lie above the ideal line. The model utilizes the principal component analysis (PCA) for dimensionality reduction, which is designed to process high-dimensional habitat factor data and pass these factors as inputs to a backpropagation neural network (BP neural network) in order to predict locust densities. The PCA removes noise and redundancy from the data and extracts the most important features, and BP neural networks are an effective non-linear regression method commonly used in complex pattern recognition and prediction problems. Overall, the model shows a good predictive ability, especially for locust density prediction in moderate ranges. However, for high-density areas, the model may need further tuning to improve prediction accuracy. A possible reason for this is that some information, such as meteorological factors with a low correlation, may be discarded during the PCA analysis.

Figure 8 below shows the validation results of the SVR (support vector regression) model. The root mean square error (RMSE) is 1.8487, which indicates an average deviation between the model’s predictions and the actual values. A lower RMSE signifies smaller prediction errors and, thus, this model’s RMSE indicates a relatively good predictive accuracy. The coefficient of determination (R²) is 0.8955, suggesting that the model accounts for a significant portion of the data’s variability. This is an improvement over the PCA-BP neural network regression model, meaning that the predictions of this model are more accurate than those of the previous one. However, the data points in certain areas of the graph show a degree of dispersion, indicating a decrease in predictive accuracy in these regions.

By observing Figure 9, which presents the results of using a BP neural network only for regression, it can be seen that the graph consists of two parts: the training process loss curve and the comparison between actual and predicted values. The validation loss (orange line) starts high and then rapidly decreases, indicating improvement in model learning during the initial phase. After several training epochs, it stabilizes, suggesting that the model achieves a lower error rate on the training data without showing signs of significant overfitting or underfitting, as the validation loss does not start increasing but instead remains consistent with the training loss. The coefficient of determination (R^2^) is 0.9158, indicating that the variability predicted by the model is highly correlated with actual data variability, and the model can explain 91.58% of the data’s variability. The root mean square error (RMSE) is 2.0178, signifying that the average deviation between the model’s predictions and actual observed values is 2.0178 units, which is a relatively small error, thus indicating high predictive accuracy. Most data points are tightly clustered around the dashed line, which represents a good prediction scenario. The distribution of data points suggests that the predictions are generally very close to actual values, especially within the middle range. However, for some lower actual values, the model’s predictions appear to be slightly worse.

The results of the deep belief network (DBN) regression model are illustrated in Figure 10. The root mean square error (RMSE) is 1.4986, indicating that, on average, the deviation between the model’s predicted values and actual values is about 1.4986 units. This relatively low RMSE value suggests that the model has high accuracy in predicting locust density. The coefficient of determination (R²) is 0.9314, meaning that the model’s predicted values explain 93.14% of the variance in the actual values, indicating strong predictive performance. The blue dots represent the actual observed values and the predicted values, and they are generally distributed along the red dashed line, demonstrating a good match between the model’s predictions and the actual situation. This indicates that the DBN model is effective in capturing the relationship between input features and locust density. While the model generally performs well, there are still some data points that deviate significantly from the ideal prediction line, and most predicted values are somewhat lower, suggesting that the model may not perfectly predict in certain scenarios. Compared with the BP neural network, the deep belief network exhibits superior predictive performance. The deep belief network is a generative model composed of multiple layers of Restricted Boltzmann Machines (RBMs), possessing powerful feature extraction capabilities. By learning the representation of data layer by layer, the DBN can capture complex structures and patterns within the data. This gives the DBN an advantage in handling highly non-linear and high-dimensional data, potentially resulting in a higher prediction accuracy in inversion regression tasks.

Figure 11 shows the comparison of the predicted values of the random forest regression model and actual values. The root mean square error (RMSE) of this model’s validation results is 1.0144, which is relatively low compared to that of the other four models previously discussed, indicating that the average deviation between the model’s predictions and actual values is only 1.0144 units. In regression models, this is a good indicator of high predictive accuracy. The coefficient of determination (R^2^) is 0.9685, a value higher than that of the other four predictive models, suggesting a very high correlation between the model’s predictions and actual data. The model can explain 96.85% of the variability in the actual data. Although there are deviations in individual samples, overall, the random forest model’s predictions are quite ideal. This demonstrates that the random forest regression method, using environmental variables, can successfully predict grassland locust density. Although the overall performance is excellent, we can see some slight deviations between several predicted points and the ideal prediction line. The incomplete accuracy of the model’s predictions may be due to some outliers or the model’s inability to fully capture all relevant factors. Among these outliers are the large errors in meteorological factors and significant errors in locust density data.

Table 8 provides a comparative analysis of the accuracy of the five models. In the table, it is evident that, among these models, the random forest regression model performs the best, followed by the deep belief network regression model. The BP neural network regression and SVR models show moderate performance, while the PCA-BP neural network regression model has relatively lower performance. Random forest and deep belief networks are more effective in handling and learning the complex non-linear relationships present in habitat factor data. Based on the comprehensive analysis above, after comparing the predictive effectiveness of these five methods on the test set, it is concluded that random forest regression is more effective in extracting features of environmental variables at grassland locust sample points, thereby making it more accurate in predicting the distribution of grassland locust density.

As shown in Figure 12, a distribution map of locust density is derived from the inversion of the trained random forest model, reflecting the distribution of locust density in the region in 2023. As can be seen in the figure, locust density in the southwest and northeast of the region is relatively high, while locust density in the middle and southeast is relatively low, which is consistent with actual survey results in previous years.

Figure 13 shows the specific error situations of the random forest model. Since the root mean square error (RMSE) of the random forest model was 1.01, all points with a difference between the predicted value and the actual value greater than 1 were considered points with large errors. A total of 229 points were detected in this figure, of which 42 points had relatively large errors, and the proportion of points with smaller errors was 82%. Moreover, through a data analysis, it was found that the error rate of the points with a locust density of 70 was relatively high, reaching 50%, and the error rate of the points with a locust density of 60 also reached 45%. The error rate of the points with a locust density of 55 was 31%, and the error rate of the points with a locust density of 40 was 46%. The errors in the inversion results of other locust densities accounted for a smaller proportion. It can be seen in the figure that these error points are approximately evenly distributed.

Figure 14 shows a comparison between the actual and predicted values of the inversion results of locust density in Xiwuzhumuqin Banner in 2023. A total of 229 sample points were verified, and the actual values were sourced from the grassland monitoring station in Xiwuzhumuqin Banner.

As shown in Figure 14, the random forest inversion model can well invert the real value curve, but there are a few points with large errors, which may be caused by input error data. It seems that the random forest model can more accurately analyze the importance of environmental variables to locust density, thus achieving higher inversion accuracy [29]. The experiment in this paper is currently limited by a small number of sampling points. In the future, we expect to incorporate data such as slope, soil type, above-ground biomass, and altitude. Among them, slope contributes significantly to egg-stage precipitation. Of course, vegetation coverage can also be included, as it also determines the occurrence of grassland locusts. All these environmental factors constitute the habitat preferences of locusts, among which surface temperature during the egg stage, NDVI, soil moisture, and nymph-stage precipitation are significant factors affecting the density of grassland locusts [30].

## 5. Conclusions

This study focuses on the core needs of early warning and monitoring of grassland locust plagues. The high-locust-infestation areas in Xiwuzhumqin Banner were selected as the research objects. A monitoring dataset for grassland locusts was constructed on the ground sample points in these areas through the comprehensive use of multi-source remote sensing data and related meteorological data. On this basis, five different regression prediction models were constructed and tested; namely, BP neural network regression based on the principal component analysis, random forest regression, BP neural network regression only, deep belief network regression, and SVR models, and their prediction effects were compared. It was found that the random forest regression model demonstrated excellent performance on the test set, with a coefficient of determination (R2) of 0.9685, a root mean square error (RMSE) of 1.0144, and the best values for all evaluation metrics compared with those of the other four regression models.

To summarize, the use of random forest regression models for grassland locust monitoring not only improves the timeliness and accuracy of early warning but also enables the real-time, dynamic, and wide-area monitoring of locust infestations. This is crucial for taking precise preventive and control measures in risky areas, greatly reducing economic losses, as well as ensuring food security. Subsequently, relevant analyses of climate patterns, topography, vegetation cover, and many other factors can be added to more accurately predict locust plague information.

## Figures and Tables

**Figure 1 sensors-24-03121-f001:**
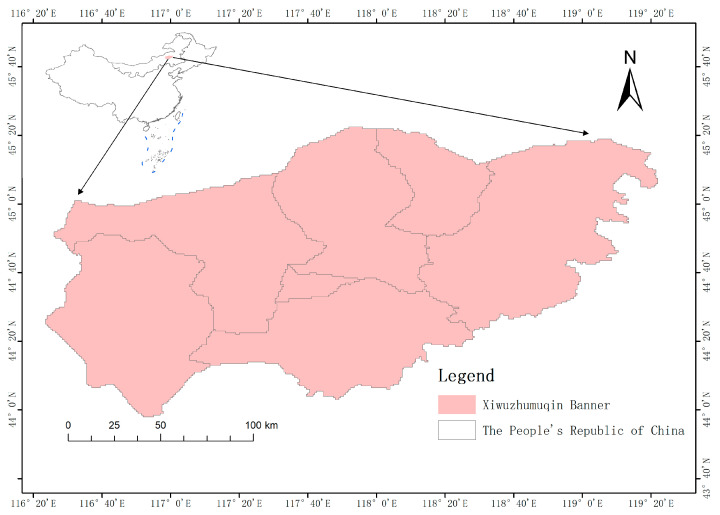
Map of the Study Area.

**Figure 2 sensors-24-03121-f002:**
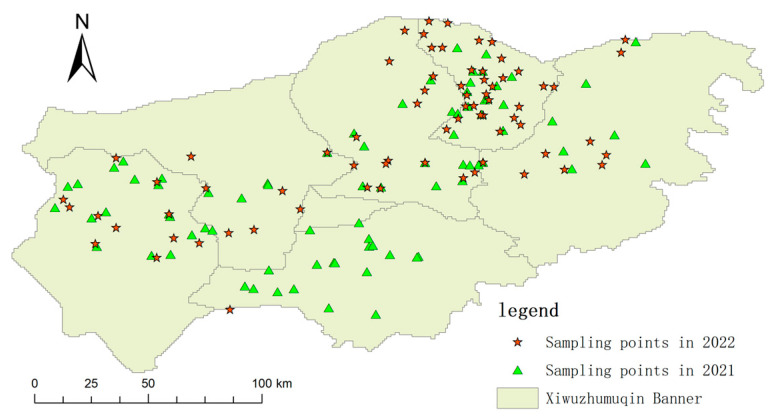
Map of Locust Survey Points in 2021 and 2022.

**Figure 3 sensors-24-03121-f003:**
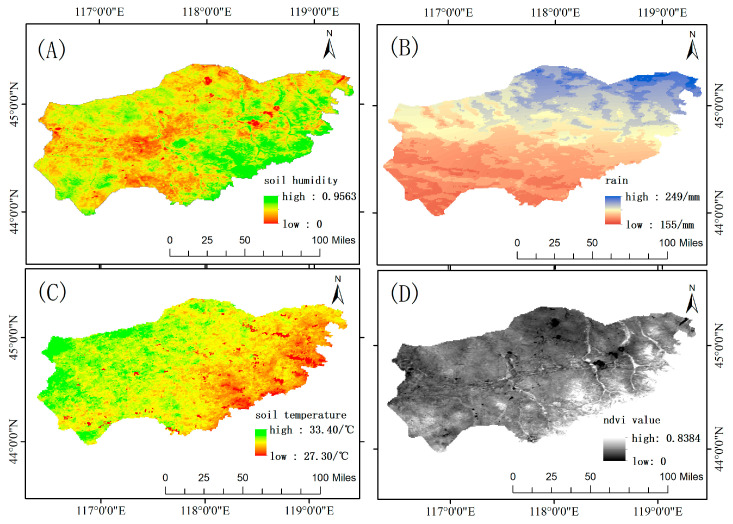
Remote sensing data: (**A**) soil moisture data; (**B**) precipitation data; (**C**) land surface temperature data; and (**D**) NDVI data (Normalized Difference Vegetation Index data).

**Figure 4 sensors-24-03121-f004:**
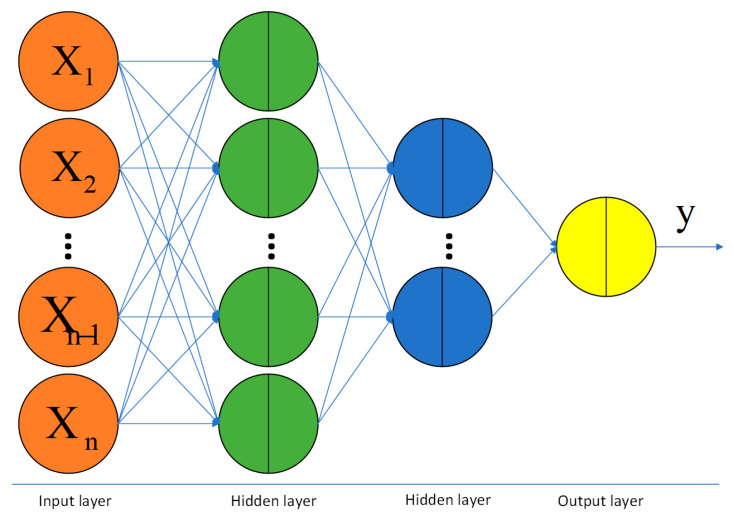
Schematic Diagram of the BP Neural Network Model.

**Figure 5 sensors-24-03121-f005:**
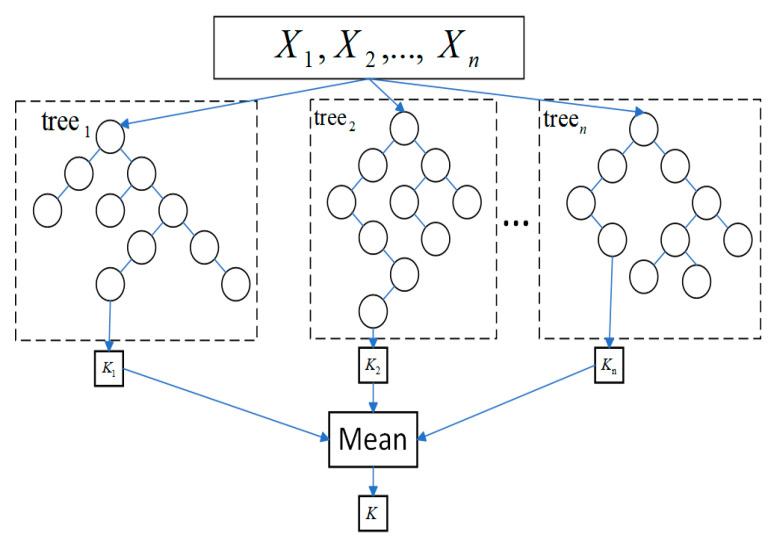
Schematic Diagram of the Random Forest Regression Model.

**Figure 6 sensors-24-03121-f006:**
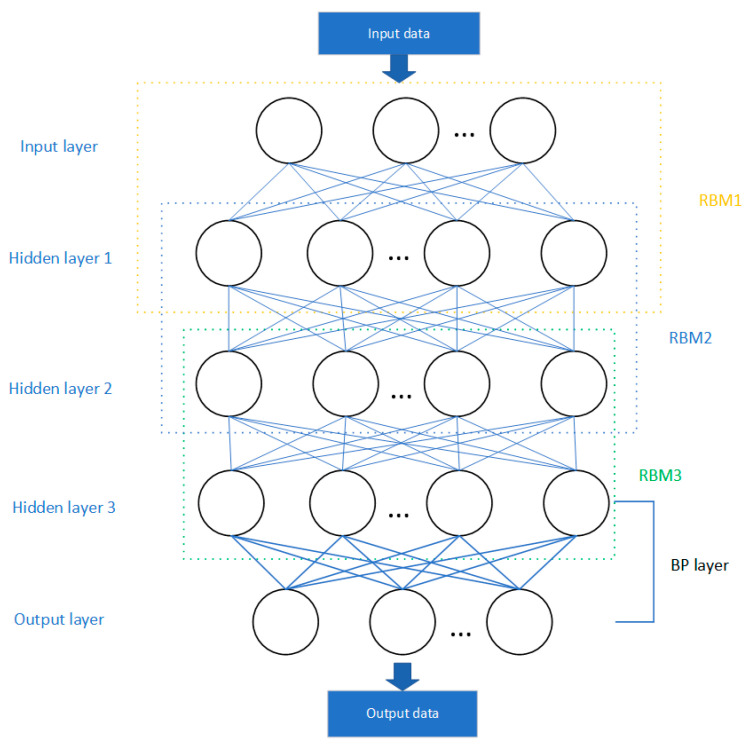
Schematic Diagram of the Deep Belief Network Regression Model.

**Figure 7 sensors-24-03121-f007:**
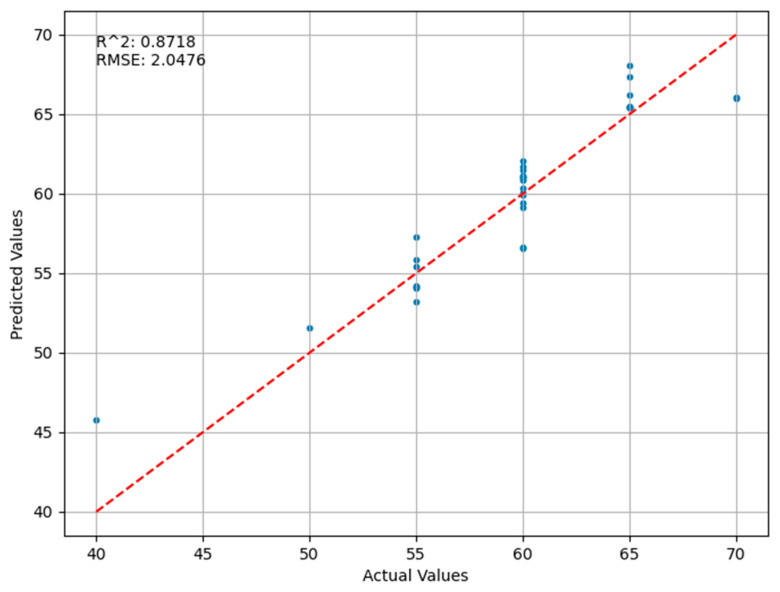
Comparison of Predicted and Actual Values in the PCA-BP Neural Network Regression Model.

**Figure 8 sensors-24-03121-f008:**
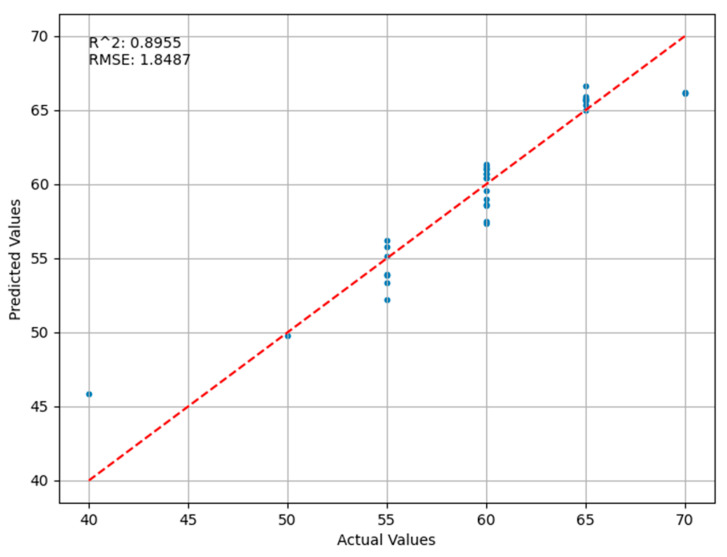
Comparison of Predicted and Actual Values in the SVR Regression Model.

**Figure 9 sensors-24-03121-f009:**
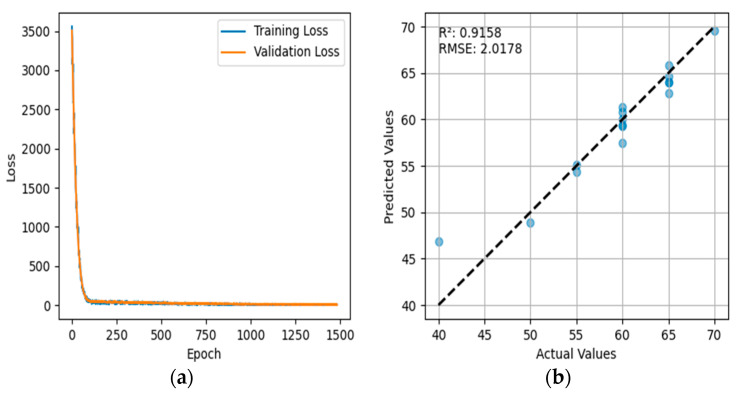
Results of the Backpropagation Neural Network: (**a**) variation of the backpropagation loss function; (**b**) comparison between predicted values and actual values in the Backpropagation Neural Network.

**Figure 10 sensors-24-03121-f010:**
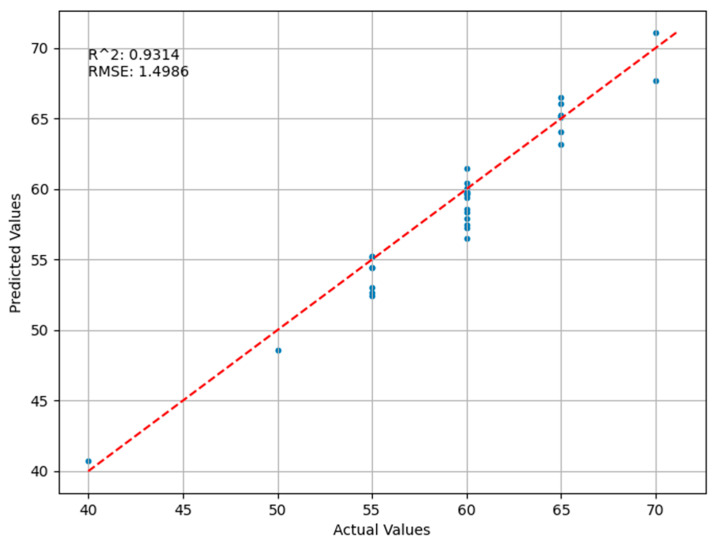
Comparison of Predicted and Actual Values in the Deep Belief Network Regression Model.

**Figure 11 sensors-24-03121-f011:**
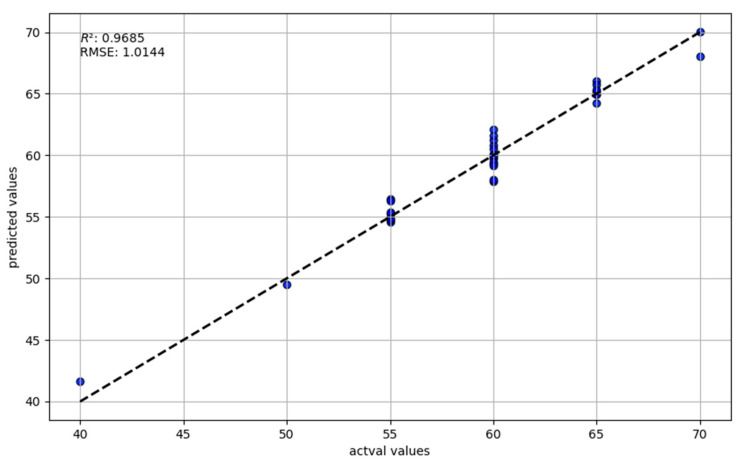
Comparison of Predicted and Actual Values in the Random Forest Regression Model.

**Figure 12 sensors-24-03121-f012:**
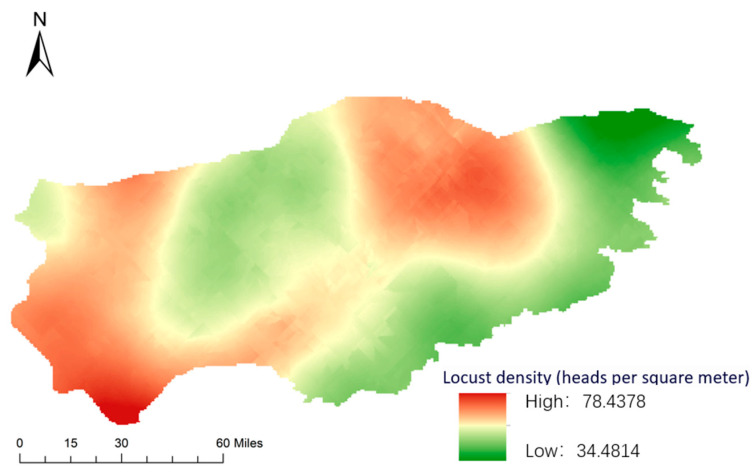
The inversion map of locust density in Xiwuzhumuqin Banner in 2023.

**Figure 13 sensors-24-03121-f013:**
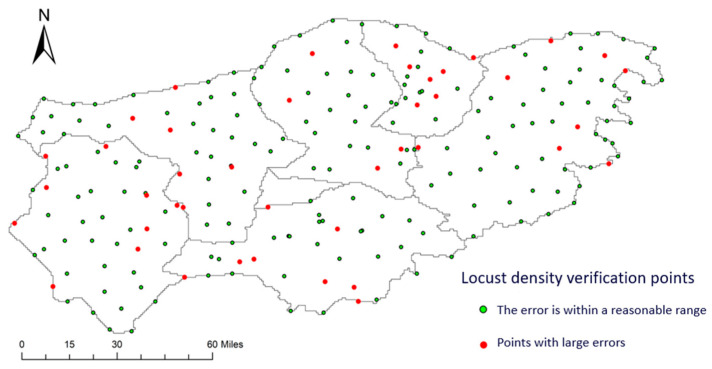
Distribution map of locust density inversion errors.

**Figure 14 sensors-24-03121-f014:**
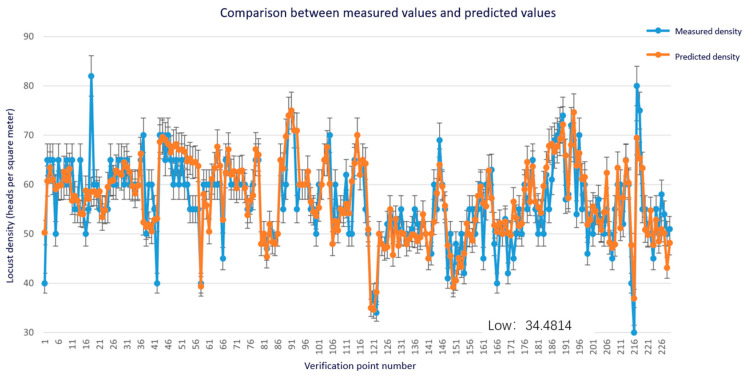
Fitting curve of predicted values and actual values.

**Table 1 sensors-24-03121-t001:** Dataset Acquisition Schedule.

Dataset	Time Period	Acquisition Time	Time Resolution	Spatial Resolution
Grassland Locust Density Data	2021, 2022	10 April 2023	One year	$1\;{km}^{2}$
Meteorological Data	From 1 January 2020 to 30 December 2022	12 April 2023	A ten-day period	$1\;{km}^{2}$
The daily 1 km all-weather land surface temperature dataset of China’s mainland and surrounding areas	From 1 January 2020 to 30 December 2022	15 April 2023	Daily	$1\;{km}^{2}$
Daily all-weather surface soil moisture data set with 1 km resolution in China (2003–2022).	From 1 January 2020 to 30 December 2022	20 April 2023	Daily	$1\;{km}^{2}$
The Monthly Precipitation Dataset of China with a Resolution of 1 km under Multiple Scenarios and Modes for 2021–2100	From 1 January 2020 to 30 December 2022	22 April 2023	Monthly	$1{km}^{2}$
NDVI	From 1 January 2020 to 30 December 2022	28 April 2023	Eight days	$1{km}^{2}$

**Table 2 sensors-24-03121-t002:** The correlation between locust density and 10-day average (daytime/night-time) land surface temperature.

Correlation Parameter	Correlation	Significance
The average daytime land surface temperature in late February 2021	−0.549 *	0.033
The average daytime land surface temperature in early April 2021	−0.664 **	0.007
The average daytime land surface temperature in mid-May 2021	−0.637 *	0.016
The average night-time land surface temperature in mid-August 2020	−0.545 *	0.030
The average night-time land surface temperature in mid-January 2021	0.554 *	0.038

“**” indicates significance at the 0.01 level (two-tailed); “*” indicates significance at the 0.05 level (two-tailed).

**Table 3 sensors-24-03121-t003:** The correlation between locust density and average precipitation in 10-day periods.

Correlation Parameter	Correlation	Significance
The average precipitation in early December 2020	−0.613 *	0.041
The average precipitation in early April 2021	0.652 *	0.033
The average precipitation in mid-June 2021	0.534 *	0.027

“*” indicates significance at the 0.05 level (two-tailed).

**Table 4 sensors-24-03121-t004:** The correlation between locust density and average soil moisture in 10-day periods.

Correlation Parameter	Correlation	Significance
The average soil moisture in early July 2020	0.625 *	0.015
The average soil moisture in mid-October 2020	−0.742 **	0.003
The average soil moisture in mid-April 2021	−0.402 *	0.040

“**” indicates significance at the 0.01 level (two-tailed); “*” indicates significance at the 0.05 level (two-tailed).

**Table 5 sensors-24-03121-t005:** The correlation between locust density and average NDVI in 10-day periods.

Correlation Parameter	Correlation	Significance
The average NDVI in mid-August 2020	−0.471 *	0.047
The average NDVI in mid-May 2021	0.422 *	0.045

“*” indicates significance at the 0.05 level (two-tailed).

**Table 6 sensors-24-03121-t006:** The random forest-gain importance score of environmental factor variables relative to locust density.

Time	Average Daytime Land Surface Temperature	Average Night-time Land Surface Temperature	Average Precipitation	Average Soil Moisture	Average NDVI
The first 10 days of July 2020	0.03702	0.00036	0.00485	0.30258	0.00003
The middle 10 days of August 2020	0.00998	0.43022	0.00135	0.01309	0.54201
The middle 10 days of October 2020	0.01008	0.01326	0.00483	0.29548	0.00012
The first 10 days of December 2020	0.03123	0.01225	0.33402	0.01322	0.00005
The middle 10 days of January 2021	0.01958	0.45068	0.00422	0.01106	0.00006
The last 10 days of February 2021	0.19425	0.01423	0.00013	0.00039	0.00002
The first 10 days of April 2021	0.25634	0.00589	0.30748	0.00006	0.00011
The middle 10 days of April 2021	0.02584	0.01135	0.00008	0.31029	0.00001
The middle 10 days of May 2021	0.28735	0.00235	0.00469	0.00304	0.45689
The middle 10 days of June 2021	0.03043	0.01488	0.32453	0.01301	0.00013

**Table 7 sensors-24-03121-t007:** Dataset for the locust density inversion model.

Serial Number	N0_01	T_01	D1_01	D2_01	J_01	N0_02	…	HCMD
1	0.4922	0.1484	3.2903	1.0000	0.7962	0.8211	…	65
2	0.5370	0.1769	4.6667	0.8963	0.7901	1.0000	…	40
3	0.4949	0.1550	4.7378	0.9121	0.7863	1.0000	…	32
4	0.5846	0.1553	4.4478	1.0000	0.8276	0.7962	…	15
5	0.4734	0.1435	3.8977	0.8942	0.8100	0.9576	…	97
6	0.5701	0.1631	4.8254	1.0000	0.7768	0.7981	…	68
…	…	…	…	…	…	…	…	…

**Table 8 sensors-24-03121-t008:** Comparison of Model Accuracies.

MODEL	R2	RMSE
PCA-BP Neural Network Regression	0.8718	2.0476
SVR regression	0.8955	1.8487
BP neural network regression	0.9158	2.0178
Deep confidence network regression	0.9314	1.4986
Random Forest Regression	0.9685	1.0144

## Data Availability

The data presented in this study are available on request from the corresponding author.

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
