# Peer review of "Remote Sensing Monitoring of Grassland Locust Density Based on Machine Learning"

_sensors, 2024, doi:10.3390/s24103121_

Round 1

Reviewer 1 Report

Comments and Suggestions for Authors

The paper is well-organized.

I would recommend to carry out further validation and comparison with other methodologies. This could enhance the robustness of your research conclusions.

Author Response

Dear reviewer, I have revised my article to make the English language more professional and fluent.

Reviewer 2 Report

Comments and Suggestions for Authors

This study "Remote sensing monitoring of grassland locust based on machine learning" tries to describe the use of AI models for the monitoring of Locust in in Xiwuzhumqin Banner by using remote sensing datasets. 

This research has some serious fundamental flaws. I think authors need to rewrite the paper according to the technical-scientific writing. There are numerous grammatical mistakes and there are many sentences that makes no sense at all. The statistical approach of this paper is not accurate as well. Authors just use Pearson correlation for the comparison purpose which will not give any credible results. Also, references are not worthy of the credible journals like sensors. Most of the citations are of Masters degree and conferences and some websites. They should read more peer-reviewed journals then write a paper.

Some of the problems I will highlight.

Table 1 – There is no description of the table and no explanation of the headings as well. Readers can not deduce any useful information from the table. For example, what is HCMD.

Figure 2 . This flow chart does not provide any useful information. Also, the caption is irrelevant. Technology roadmap what does it mean.

Section 3.1 - 

daytime surface temperatures were acquired from publicly available high-precision datasets.” There is no reference or link to the website from where they acquired the data set.

Data Repetition

“where the locust density survey points for 2021 should contain five remote sensing data from July 2020 to July 2021. The locust density survey points for 2022 should contain 5 remote sensing data from July 2021 to July 2022.”

The language was non-technical throughout the paper. I would suggest reviewing the paper from the technical point of view and rewriting and using technical words. For example, “Data was standardized to 1 km”. instead of using standardization use “resampled to 1 km”

Almost 80 percent of the paper is like an explanation of the machine learning algorithm which is unnecessary because now these are well explained concepts and methods in scientific community. There is no need to explain every model in detail.

How the authors created the model there is no explanation for that.

Authors main aim was to create a better prediction for locust detection but they did not provide any concrete results for the region. For example, how many days before their model will predict the locust attack?

There is no table for the explanation of all the data sets.

I would suggest reading a few papers and then do the analysis again.

For example, this paper these two papers have the same conclusion that Random forest gives a better estimation of locust prediction.

https://www.mdpi.com/2076-3417/13/14/8266

Or this paper deals with the application of ML for locust prediction.

https://www.sciencedirect.com/science/article/abs/pii/S0140196321001658

I would suggest the authors read these papers and rewrite and present their findings accurately. Especially the writing style.

Comments on the Quality of English Language

English quality is really low but the major problem was not English writing it was not scientific technical writing at all. There were grammatical errors as well as some sentences did not make any sense at all. 

Author Response

Dear reviewer,

I have carefully revised my article based on your guidance and suggestions. Here are the specific changes I have made:

  1. Regarding statistical methods, I have added random forest weights to compare with the Pearson correlation for obtaining feature variables.

  2. Regarding the tables, I have provided detailed descriptions for all of them.

  3. Regarding the public datasets, I have included their detailed links.

  4. Regarding the English description, I have polished the language and made it more professional.

  5. Regarding the conclusions and analysis, I have made detailed optimizations and studied the journal articles you provided.

I am extremely grateful for your feedback, which has greatly improved my writing and academic skills. I would appreciate any further suggestions for improvement. I am looking forward to your next comments, as your expertise is highly esteemed by me.

Thank you once again for your time and effort in reviewing my work.

Sincerely,

Qiang Du

Reviewer 3 Report

Comments and Suggestions for Authors

The manuscript entitled “Remote sensing monitoring of grassland locust based on machine learning” revises the application of remote sensing technology as a new method for monitoring desert locusts. The manuscript describes and analyses several machine learning algorithms. The experimental results are interesting from a scientific point of view. The structure of the manuscript is well organized although it can be improved. Diagrams , figures and schemes correspond  properly to the context and well visualize the idea of the authors.

Recommendations

1.     The abstract should be revised. I recommend that authors organized the structure of the abstract in a way that will represent more of the essence of the manuscript. Some more details will be favorable for the readers.

2.     The conclusion should include more information that linked the title of the manuscript with the results. Avoid repeating the text in the abstract. The last paragraph in the conclusion is suitable.

3.     The authors could emphasize more on the “validation and accuracy comparison” issue.

4.     A map showing the distribution of grassland locust, including the prediction,  could be interesting.

Some points to be re-checked or clarified:

-        Do you mean “Introduction” when using “Introductorty”?

-         On p.3 in the last paragraph it is mentioned “ Another component of the multi-source remote sensing data is the 2021-2100 China…….” And “It covers the period from January 2021 to December 2100….”. Are the years correct?

-        In section 3.2. Min-Max Normalization there is some space that should be put between words.

-         On p.12 /4.2. Analysis of results/ in the sentence “The possible reason for this is that some information was discarded during the PCA analysis.” Could you please specify what kind of information was discarded?

-        On p.14 /4.2. Analysis of results/ in the sentence “Compared to the BP Neural Network, the Deep Belief Network shows superior predictive performance, indicating that its multiple layers of RBMs are more effective in extracting features from the samples.” Could you please specify the features from the samples?

-        On p.15 /4.2. Analysis of results/ in the sentence “This may indicate that for certain data points, the model's predictions are not completely accurate, possibly due to some outliers or the model's inability to capture all relevant factors completely.” Could you specify which are the certain data points?

-        Do you mean “Conclusion” when using “Conclude”?

-        I could not find references number 2,6,7,9,14,22,26. Could you please send me some information for them ?

The manuscript is relevant for the Journal Sensors MDPI after major revision.

Good luck!

Reviewer

Comments on the Quality of English Language

Author Response

Dear reviewer,

I have carefully revised my article based on your guidance and suggestions. Here are the specific changes I have made:

  1. I have created a map to demonstrate the distribution of grasshoppers, including their predicted distribution.

  2. I have corrected the typo in the title, changing "Introductorty" to "Introduction."

  3. On page 3, in the last paragraph, I have corrected the incorrect years mentioned in the text. The phrase "another component of multi-source remote sensing data is China's... from 2021 to 2100" has been revised, and the timeline "from January 2021 to December 2100" has also been corrected.

  4. On page 12, in Section 4.2 "Results Analysis," I have provided a more specific explanation for the sentence "A possible reason is that some information was discarded during the PCA analysis."

  5. On page 14, in Section 4.2 "Results Analysis," I have revised the sentence comparing the prediction performance of the deep belief network with the BP neural network. The new sentence reads: "Compared to the BP neural network, the deep belief network demonstrated superior prediction performance, indicating its multilayer RBMs are more effective in extracting features from the samples."

  6. On page 15, in Section 4.2 "Results Analysis," I have made modifications to the sentence discussing the accuracy of the model's predictions for certain data points. The revised sentence reads: "This may suggest that for some data points, the model's predictions are not entirely accurate, potentially due to outliers or the model's inability to fully capture all relevant factors."

  7. I have corrected the typo in the conclusion section, changing "Conclude" to "Conclusion."

  8. Regarding the references, numbers 2, 6, 7, 9, 14, 22, and 26 are domestic Chinese journals that may not be accessible to you. Please let me know if you would like me to replace these references with more internationally accessible ones.

  9. I have made significant revisions to the article, correcting English errors and improving the argumentation.

I am extremely grateful for your feedback, which has greatly improved my writing and academic skills. I would appreciate any further suggestions for improvement. I am looking forward to your next comments, as your expertise is highly esteemed by me.

Thank you once again for your time and effort in reviewing my work.

Sincerely,

Qiang Du

Round 2

Reviewer 2 Report

Comments and Suggestions for Authors

1-     The Response provided by the authors is not enough. They should answer every question raised by reviewers separately.

2-     Major problem of this paper is again English language. Even though they have corrected many parts of the paper but still it has some serious flaws.

3-     Overall structure of the paper is not correct and very confusing as well. Authors have done a very hard work but that is not presented in the accurate and readable manner. Importantly headings are confusing as well.

4-     Figure 1 – Include China Map - Study area map should be made again. This is an international journal. They should show complete China map and then show where the study conducted. It is not easily understandable where actually the study takes place while looking at the map.

5-     Incorrect caption of the all the figures. They are showing study area map but they captioned it schematic diagram of the area. This is not the schematic map or a diagram. Its just a study area map. Authors should at least know the difference.

6-     It is still not understandable how they created the locust sampling data. Authors should explain in detail more what is meant by “data is sourced from Xiwuzhumqin Banner Grassland Workstation” what is this workstation. If they have calculated by manually counted the locust or by capturing locust and then sample it. Then it should be clear and precise   I personally believe it is the writing style that is confusing the paper. For example, they have written “The study obtained data on locust” “They should write. During the field survey locust data was collected by “

7-     In section 2.3 – How they thoroughly corrected the meteorological datasets. What methods they have adopted. Do they double check through the satellite-based datasets.  Also, this sentence does not make any sense “Additionally, it includes detailed longitude and latitude coordinates of the meteorological stations.” What do they mean by detailed latitude and longitude.

8-     Figure 3 caption is again incorrect. What is partial remote sensing data. This is misinformation. They should also make table of each dataset along with their date of acquisition. I have previously asked for this but they did not incorporate.

9-     Table 1 - Instead of writing current year they should write the full date of the year.

10- Table 5 – What is “The random forest-gain importance score” what is this “gain” Explain this gain in the text above.

11- This paper lacks a discussion portion. Which is crucial part of any scientific paper. Authors need to make a separate section of Discussion and they should explain different studies and compare their results with their outcome. They showed scientifically discuss why this method is good for locust detection as compared to other methods and what benefit it can provide. Is it ok to repeat the experiment in other parts of the world and what different paraments should also by consider and what limitations they faced.

12- References provided in this paper is not worthy of a scientific journal. They have given multiple Masters degree references. They should provide their published results instead. Also the formatting of the references is not aligned with the journal guidelines. 

Comments on the Quality of English Language

The quality of English is really low. Although they have improved a lot from the previous attempt but still meaning is lost while reading. 

Reviewer 3 Report

Comments and Suggestions for Authors

Dear  authors,

Thank you for considering my recommendations.

Regards,

Reviewer

Author Response

Comments 1: Improve English level.

Response 1: I have used the English editing service of MDPI to improve the English expression level of the article.

Round 3

Reviewer 2 Report

Comments and Suggestions for Authors

Authors have significantly improved the paper but I have some minor corrections to suggest. I am reviewing this paper for the third time. Authors have incorporated almost all the comments I have asked them to make. 

They should remove the “Research on” from the title and just start with the “Remote Sensing based monitoring of Grassland Locust Density using machine learning techniques”

Please correct your reference number 8, 9 and 10 these are not correctly formatted as APA style or MDPI style format as other references it seems authors have manually entered these references. 

Comments on the Quality of English Language

There are few minor English corrections still needs to be done but overall it has improved a lot. 
